

# The forward-backward asymmetry and differences of partial moments in inclusive semileptonic $B$ decays

**Florian Herren**

Fermi National Accelerator Laboratory, Batavia, IL, 60510, USA

⋆ florian.s.herren@gmail.com

## Abstract

Global fits to moments of kinematic distributions measured in inclusive semileptonic $B \to X_c l \nu_l$ enable the determination of the Cabibbo-Kobayashi-Maskawa matrix element $|V_{cb}|$ together with non-perturbative matrix elements of the heavy quark expansion. In current fits, only two distinct kinematic distributions are employed and, as a consequence, higher moments of these distributions need to be taken into account to extract the relevant non-perturbative matrix elements. The moments of a given distribution are highly correlated and experimental uncertainties increase for higher moments. To address these issues, Turczyk suggested the inclusion of the charged lepton forward-backward asymmetry $\mathcal{A}_{FB}$ in global fits, since it provides information on non-perturbative parameters beyond the commonly used moments. It is possible to construct differences of partial moments of kinematic distributions, which can provide additional information on the non-perturbative parameters beyond $\mathcal{A}_{FB}$ and are studied in this work for the first time. Further, experimental cuts on the four-momentum transfer square are studied and are shown to preserve the shape of the angular distribution, in contrast to commonly used cuts on the lepton energy. Finally, the impact of final-state radiation and experimental lepton identification requirements on measurements of $\mathcal{A}_{FB}$ and differences of partial moments are discussed.



# 1   Introduction

Semileptonic $B$ decays allow for the determination of the magnitude of the Cabibbo-Kobayashi-Maskawa (CKM) matrix element $|V_{cb}|$. Inclusive decays, which take into account all possible final states involving a charmed meson, can be reliably described by the Heavy Quark Expansion (HQE), a double expansion in the inverse mass of the bottom quark $1/m_b$ and the strong coupling constant, $\alpha_s$. See, e.g., [1].

In this framework, the total rate and moments of kinematic distributions for $B \to X_c l \nu_l$ decays, where $X_c$ denotes all possible hadronic final states involving a charmed meson, has been computed up to $\mathcal{O}(\alpha_s^3)$ at $\mathcal{O}(1/m_b^0)$ [2–7], $\mathcal{O}(\alpha_s)$ at $\mathcal{O}(1/m_b^2)$, [8–11] and tree-level for contributions at $\mathcal{O}(1/m_b^3)$ [12],[1] $\mathcal{O}(1/m_b^4)$ [13] and $\mathcal{O}(1/m_b^5)$ [14].[2] The expansion in $1/m_b$ involves hadronic matrix elements that cannot be computed in a perturbative manner: two at $\mathcal{O}(1/m_b^2)$, two at $\mathcal{O}(1/m_b^3)$, nine at $\mathcal{O}(1/m_b^4)$ and eighteen at $\mathcal{O}(1/m_b^5)$. To extract $|V_{cb}|$ from measurements of the decay rate, these non-perturbative parameters must be determined from additional measurements. To this end, moments of kinematic distributions, such as the hadronic-mass moments $\langle M_X^n \rangle$ [15–20] and lepton-energy moments $\langle E_l^n \rangle$ [16, 19, 21, 22] are measured. These moments can also be predicted in the HQE and corrections at similar accuracy as for the total rate are available [4, 5]. The decay rate and moments are then employed in a global fit to extract $|V_{cb}|$ in conjunction with the non-perturbative parameters [23–27]. Inclusive determinations of $|V_{cb}|$ reveal tensions with determinations of $|V_{cb}|$ in exclusive decays, i.e. decays in which only a specific charmed hadron is taken into account, of $3\sigma$ [28]. The non-perturbative parameters and $|V_{cb}|$ enter predictions for processes sensitive to new physics contributions, such as $B \to X_s \gamma$ or $B \to X_s ll$. Consequently, a precise knowledge of these parameters is paramount to constrain physics beyond the Standard Model. The two main weaknesses of the inclusive decay approach are the strong correlations among moments of a given distribution and the large number of non-perturbative parameters at higher orders in the HQE.

A possibility to alleviate the second of these weaknesses is to study observables that depend on a smaller set of non-perturbative parameters. To this end, moments of the four-momentum transfer square $\langle (q^2)^n \rangle$ have been suggested as an alternative possibility to extract $|V_{cb}|$ [29]. The $\langle (q^2)^n \rangle$, like the total semileptonic decay rate, obey a symmetry of the HQE known as

---

[1]With the exception of moments of the $q^2$-distribution, which are known at $\mathcal{O}(\alpha_s)$ [11].

[2]Terms going like $1/m_b$ vanish [1].

reparametrization invariance (RPI), allowing for a partial resummation of the HQE [30]. As a consequence, they depend on fixed linear combinations of subsets of the non-perturbative parameters, eight through $\mathcal{O}\left(1/m_b^4\right)$, compared to thirteen for the lepton-energy and hadronic-mass moments. The reduction in the number of fit parameters allows for a fit only taking into account the $\left\langle\left(q^2\right)^n\right\rangle$ and the total semileptonic decay rate. Recently, the $\left\langle\left(q^2\right)^n\right\rangle$ for $n \leq 4$ have been measured for the first time by the Belle experiment [31] and a preliminary determination of $|V_{cb}|$ has been presented in Ref. [32].

The first weakness can be alleviated by adding complementary observables that are less correlated with existing ones. Further, knowledge of additional observables allows to over-constrain global fits and test the fits for internal consistency between different observables. In this light, Turczyk suggested including the charged lepton forward-backward asymmetry, $\mathcal{A}_{FB}$, in inclusive semileptonic $B \to X_c l \, \nu_l$ decays [33]. The asymmetry is defined by

$$\mathcal{A}_{FB} = \frac{1}{\Gamma}\left(\int_{-1}^0 \mathrm{d}z \frac{\mathrm{d}\Gamma}{\mathrm{d}z} - \int_0^1 \mathrm{d}z \frac{\mathrm{d}\Gamma}{\mathrm{d}z}\right), \tag{1}$$

where $\Gamma$ is the partial decay width of semileptonic $B \to X_c l \, \nu_l$ decays and $z$ is given by the angle between the charged lepton and the $B$ meson in the rest frame of the lepton-neutrino–system, $\theta_{Bl}^q$, through $z = \cos\theta_{Bl}^q$.

A related collection of observables it the sums and differences of partial moments of the angular distribution, $\langle z^n\rangle_\pm$, which are defined as:

$$\langle z^n\rangle_\pm = \frac{1}{\Gamma}\left(\int_{-1}^0 \mathrm{d}z \, z^n \frac{\mathrm{d}\Gamma}{\mathrm{d}z} \pm \int_0^1 \mathrm{d}z \, z^n \frac{\mathrm{d}\Gamma}{\mathrm{d}z}\right), \tag{2}$$

where $\frac{\mathrm{d}\Gamma}{\mathrm{d}z}$ is the differential decay rate. Note that $\mathcal{A}_{FB} = \left\langle z^0\right\rangle_-$ and $\left\langle z^0\right\rangle_+ = 1$. In Ref. [33] it was argued that the partial moments $\langle z^n\rangle_\pm|_{n\geq 1}$ do not provide additional information on the non-perturbative parameters in the HQE.

Additionally, Turczyk suggested studying of *differences* of partial moments of observables in the forward and backward directions. For an observable $\mathcal{O}$, the sums and differences of partial moments of differential decay rates can be defined as

$$\langle\mathcal{O}^n\rangle_\pm = \frac{1}{\Gamma}\left(\int_{-1}^0 \mathrm{d}z \int \mathrm{d}\mathcal{O} \, \mathcal{O}^n \frac{\mathrm{d}\Gamma}{\mathrm{d}z\mathrm{d}\mathcal{O}} \pm \int_0^1 \mathrm{d}z \int \mathrm{d}\mathcal{O} \, \mathcal{O}^n \frac{\mathrm{d}\Gamma}{\mathrm{d}z\mathrm{d}\mathcal{O}}\right). \tag{3}$$

Although the sums $\langle\mathcal{O}^n\rangle_+$ correspond to the previously discussed moments already measured by experiments, the differences $\langle\mathcal{O}^n\rangle_-$ have not yet been studied in the literature.

Therefore, in this paper, the discussion of Ref. [33] is revisited and extended to

i) explore the impact of a cut on $q^2$ instead of the lepton energy $E_l$ in measurements of $\mathrm{d}\Gamma/\mathrm{d}z$,

ii) re-assess the value of moments of the angular distribution, $\langle z^n\rangle_\pm$,

iii) derive expressions for differences of partial moments, $\langle\mathcal{O}^n\rangle_-$, for $\mathcal{O} = \{M_X^2, E_l, q^2\}$,

iv) study the effect of final-state radiation on the angular distribution, and

v) investigate the differences between decays to muons and electrons due to experimental particle-identification requirements.

In addition to aiding the extraction of non-perturbative parameters in the HQE, the forward-backward asymmetry provides a unique opportunity to test lepton flavor universality in semileptonic $B$ decays. Not only can $\mathcal{A}_{FB}$ be measured separately for electron and muon

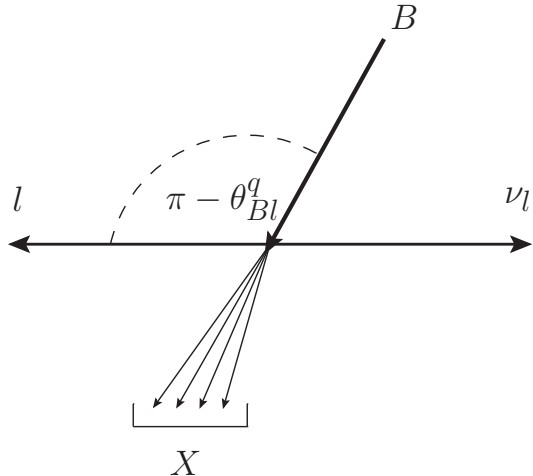

Figure 1: The decay of a $B$ meson into a lepton $l$, a neutrino $\nu_l$ and a hadronic system $X$ in the lepton-neutrino rest frame. The angle between the flight direction of lepton and $B$ meson is $\theta_{Bl}^q$ with $z = \cos \theta_{Bl}^q$.

modes, but also their difference, $\mathcal{A}_{FB}^{(\mu)} - \mathcal{A}_{FB}^{(e)}$, can be determined precisely, including correlations between the two modes. A recent study of the full Belle dataset on the angular spectrum of the lepton in $B \to D^* l \nu_l$ decays [34] revealed tensions with the Standard-Model prediction of $2\sigma$ for $\mathcal{A}_{FB}^{(\mu)}$ and $4\sigma$ for $\mathcal{A}_{FB}^{(\mu)} - \mathcal{A}_{FB}^{(e)}$ [35]. A measurement of $\mathcal{A}_{FB}$ in inclusive decays would allow for a valuable independent check of this tension.

The remainder of this paper is organized as follows. In Sec. 2 the triple differential decay rate for semileptonic $B \to X l \nu_l$ decays is derived and analytic expressions for $\mathcal{A}_{FB}$ and the aforementioned moments are presented. Next, in Sec. 3 the dependence of $\mathcal{A}_{FB}$ and the moments on the non-perturbative parameters and a cut on $q^2$ is studied. The effects of final-state radiation and particle-identification requirements on $\mathcal{A}_{FB}$ at Belle II are discussed in Sec. 4. Finally, in Sec. 5 the results of these studies are summarized and their potential impact on future determinations of $|V_{cb}|$ and non-perturbative HQE parameters is discussed.

## 2 The triple differential decay rate

Analyses of collision data taken at $B$-factories that employ the technique of hadronic tagging, such as Belle II, have access to the four-momentum of the signal $B$ meson, $p_B^\mu$. To this end, events are selected in which the decay of one of the $B$ mesons is fully reconstructed. The momenta of the reconstructed $B$ meson and of the beams are then combined to obtain $p_B^\mu$. Additionally, for semileptonic decay processes, the momentum of the signal lepton, $p_l^\mu$, and the sum of momenta of hadronic daughter particles, $p_X^\mu$, are reconstructed, thus giving access to the neutrino four-momentum:

$$ p_{\nu_l} = p_B^\mu - p_l - p_X^\mu. \tag{4} $$

To derive the differential decay rate, $z$ needs to be expressed in terms of Lorentz invariant quantities. To this end, the four-momenta of the $B$ meson, the lepton and the neutrino in the lepton-neutrino rest frame are parametrized as shown in Fig. 1.

The cosine of the angle between the flight directions of the lepton and the $B$ meson can be

written in terms of energies and momenta in the lepton-neutrino rest frame as

$$z = \frac{p_B \cdot p_{\nu_l} - p_B \cdot p_l}{\sqrt{q^2}\sqrt{(E_B)^2 - m_B^2}}, \tag{5}$$

where $q^\mu = p_{\nu_l} + p_l$ is the four-momentum transfer. Equation (5) can be re-expressed in terms of Lorentz invariant quantities as:

$$z = \frac{v \cdot p_{\nu_l} - v \cdot p_l}{\sqrt{(v \cdot q)^2 - q^2}}, \tag{6}$$

where $v^\mu = p_B^\mu/m_B$ is the velocity of the $B$ meson.

For experimental analyses, however, it is more convenient to work in the $B$-meson rest frame, in which case $v^\mu = (1,0,0,0)$ and the expression for $z$ becomes:

$$z = \frac{E_{\nu_l} - E_l}{\sqrt{(E_{\nu_l} + E_l)^2 - q^2}}. \tag{7}$$

Following [1] the triple differential decay rate in the $B$-meson rest frame can be expressed as

$$\frac{d^3\Gamma}{dz\,dq^2\,dv \cdot q} = \int \frac{d\Pi_{l\nu_l}}{dz\,dq^2\,dv \cdot q} \sum_X \sum_{s_l} \frac{\left|\langle Xl\nu_l|H_W|\bar B\rangle\right|^2}{2m_B}(2\pi)^4\delta^4\left(p_B - p_l - p_{\nu_l} - p_X\right), \tag{8}$$

where the sums are over all possible hadronic final states, $X$, and the lepton spins, $s_l$. The lepton-neutrino phase-space is given by

$$\int \frac{d\Pi_{l\nu_l}}{dz\,dq^2\,dv \cdot q} = \int \frac{d^3p_l}{(2\pi)^3 2E_l} \int \frac{d^3p_{\nu_l}}{(2\pi)^3 2E_{\nu_l}} \delta\left(q^2 - (p_l + p_{\nu_l})^2\right)$$

$$\otimes\, \delta\left(v \cdot q - E_{\nu_l} - E_l\right)\delta\left(z - \frac{E_{\nu_l} - E_l}{\sqrt{(E_l + E_{\nu_l})^2 - (p_l + p_{\nu_l})^2}}\right), \tag{9}$$

and in the Standard Model (SM) the effective weak Hamiltonian governing semileptonic $b \to c, u$ transitions takes the form

$$H_W = \frac{4G_F}{\sqrt{2}}\left[V_{cb}\left(\bar c\gamma^\mu P_L b\right)\left(\bar l\gamma_\mu P_L \nu_l\right) + V_{ub}\left(\bar u\gamma^\mu P_L b\right)\left(\bar l\gamma_\mu P_L \nu_l\right)\right]$$

$$= \frac{4G_F}{\sqrt{2}}\left[V_{cb}J_c^\mu J_{l,\mu} + V_{ub}J_u^\mu J_{l,\mu}\right]. \tag{10}$$

Although, only $b \to c$ decays are discussed explicitly in the following, the results also apply to $b \to u$ transitions since the derivation does not depend on the flavor of the daughter quark.

## 2.1 Simplification of the decay rate

Neglecting higher-order electroweak corrections, the matrix element appearing in Eq. (8) can be factored into a leptonic tensor $L^{\mu\nu}$ and a hadronic tensor $W^{\mu\nu}$ [1]:

$$L^{\mu\nu} = \sum_{s_l}\langle 0|J_l^{\nu,\dagger}|l\bar\nu_l\rangle\langle l\bar\nu_l|J_l^\mu|0\rangle$$

$$= 2\left(p_l^\mu p_{\nu_l}^\nu + p_l^\nu p_{\nu_l}^\mu - g^{\mu\nu}p_l \cdot p_{\nu_l} - i\epsilon^{\mu\nu\rho\sigma}p_{l,\rho}p_{\nu_l,\sigma}\right), \tag{11}$$

$$W^{\mu\nu} = \frac{(2\pi)^3}{2m_B}\sum_X\langle\bar B|J_q^{\nu,\dagger}|X_x\rangle\langle X|J_q^\mu|\bar B\rangle\delta^4(p_B - q - p_X)$$

$$= -g^{\mu\nu}W_1 + v^\mu v^\nu W_2 - i\epsilon^{\mu\nu\rho\sigma}v_\rho q_\sigma W_3 + q_\mu q_\nu W_4 + (v_\mu q_\nu + v_\nu q_\mu)W_5. \tag{12}$$

Here the $W_i$ are scalar structure functions which are discussed in section 2.2. In the limit of vanishing lepton mass, the contraction of leptonic and hadronic tensors is given by

$$L^{\mu\nu}W_{\mu\nu} = 2q^2 W_1 + (1-z^2)((v\cdot q)^2 - q^2)W_2 - 2q^2 z\sqrt{(v\cdot q)^2 - q^2}W_3. \tag{13}$$

Since the hadronic and leptonic currents factorize, the structure functions can only depend on the four-momentum transfer and not the individual momenta of the leptons. Thus, all integrations in Eq. (9) can be performed, leading to:

$$\int \frac{d\Pi_{l\nu_l}}{dz\,dq^2\,dv\cdot q} = \frac{1}{8(2\pi)^4}\sqrt{(v\cdot q)^2 - q^2}. \tag{14}$$

Combining Eqs. (8), (13) and (14), the triple differential decay rate is given by

$$\frac{d^3\Gamma}{dz\,dq^2\,dv\cdot q} = \frac{G_F^2|V_{cb}|^2}{8\pi^3}\sqrt{(v\cdot q)^2 - q^2}\bigg(2q^2 W_1 + (1-z^2)((v\cdot q)^2 - q^2)W_2$$
$$- 2q^2 z\sqrt{(v\cdot q)^2 - q^2}W_3\bigg). \tag{15}$$

This expression differs from the one given in Ref. [33] by a minus sign in the last term but agrees with Ref. [1] after a change of variables.

## 2.2 The hadronic tensor

The hadronic tensor $W$ is related to the time-ordered product [13]

$$T^{\mu\nu} = -i\int d^4 x\, e^{-ix\cdot(m_b v - q)}\frac{\left\langle B(p)\left|T\left[J_q^{\mu,\dagger}(x)J_q^\nu(0)\right]\right|B(p)\right\rangle}{2m_B} \tag{16}$$

by the optical theorem:

$$-\frac{1}{\pi}\text{Im}\,T^{\mu\nu} = W^{\mu\nu}. \tag{17}$$

As in the case of $W$, $T$ can be decomposed into tensor structures and scalar functions, allowing to relate each structure function $W_i$ to a scalar function $T_i$:

$$T^{\mu\nu} = -g^{\mu\nu}T_1 + v^\mu v^\nu T_2 - i\epsilon^{\mu\nu\rho\sigma}v_\rho q_\sigma T_3 + q_\mu q_\nu T_4 + (v_\mu q_\nu + v_\nu q_\mu)T_5. \tag{18}$$

The $T_i$ can be computed by an operator product expansion (OPE) of the time ordered product, leading to an expansion in $1/m_b$. In Refs. [13,14] this expansion was done at leading order in $\alpha_s$ and through $\mathcal{O}(1/m_b^4)$ and $\mathcal{O}(1/m_b^5)$, respectively, using the background-field method along the lines of Ref. [36]. In this formalism, the charm quark interacts with the background fields of the soft gluons of the $B$-meson and its propagator takes the form [37]

$$iS_{\text{BGF}} = \frac{1}{\slashed{Q} + i\slashed{D} - m_c}. \tag{19}$$

Here $q^\mu$ is re-expressed as $Q^\mu = m_b v^\mu - q^\mu$ to simplify the calculation and $D^\mu$ denotes the derivative with respect to the background gauge field. Employing the background-field propagator, the tensor $T$ can be written as [13]

$$T^{\mu\nu} = \left\langle B(p)\left|\bar{b}_v \Gamma^\mu S_{\text{BGF}}\Gamma^{\dagger,\nu}b_v\right|B(p)\right\rangle, \tag{20}$$

where $b_v$ is related to the $b$-quark field by

$$b(x) = e^{-im_b v \cdot x} b_v(x). \tag{21}$$

To perform the expansion in $1/m_b$, the background-field propagator is expanded in the covariant derivative according to [13]

$$iS_{\text{BGF}} = \frac{1}{\not{Q} - m_c} - \frac{1}{\not{Q} - m_c}(i\not{D})\frac{1}{\not{Q} - m_c} + \frac{1}{\not{Q} - m_c}(i\not{D})\frac{1}{\not{Q} - m_c}(i\not{D})\frac{1}{\not{Q} - m_c} + \cdots \tag{22}$$

and numerator structures are subsequently simplified. The resulting expressions can be matched onto the structures in Eq. (18). As a consequence, the $T_i$ depend on the parameters and kinematic quantities $m_b$, $m_c$, $q^2$, $v \cdot q$, and the charm quark propagator

$$\Delta_0 = ((m_b v - q)^2 - m_c^2 + i\epsilon). \tag{23}$$

They also, of course, depend on hadronic matrix elements that encode the non-perturbative physics. While there are no corrections and thus no matrix elements, at $\mathcal{O}(1/m_b)$, there are two relevant matrix elements at $\mathcal{O}(1/m_b^2)$,

$$\hat{\mu}_\pi^2 = -\frac{\langle B(p) | \bar{b}_v (iv \cdot D)^2 b_v | B(p) \rangle}{2M_B}, \tag{24}$$

$$\hat{\mu}_{\text{G}}^2 = \frac{\langle B(p) | \bar{b}_v (iD^\mu)(iD^\nu)(-i\sigma^{\mu\nu}) b_v | B(p) \rangle}{2M_B}, \tag{25}$$

and another two at $\mathcal{O}(1/m_b^3)$:

$$\hat{\rho}_{\text{D}}^3 = \frac{\langle B(p) | \bar{b}_v (iD^\mu)(iv \cdot D)(iD_\mu) b_v | B(p) \rangle}{2M_B}, \tag{26}$$

$$\hat{\rho}_{\text{LS}}^3 = \frac{\langle B(p) | \bar{b}_v (iD^\mu)(iv \cdot D)(iD^\nu)(-i\sigma^{\mu\nu}) b_v | B(p) \rangle}{2M_B}. \tag{27}$$

In Eqs. (25) - (27), the parameters $\hat{\mu}_\pi^2$, $\hat{\mu}_G^2$, $\hat{\rho}_{\text{D}}^3$ and $\hat{\rho}_{\text{LS}}^3$ are the kinetic energy parameter, the chromomagnetic moment, the Darwin term and the spin-orbit term, respectively. These non-perturbative parameters only depend on the decaying $B$-meson, not on the final state hadrons. Consequentely, non-perturbative parameters determined in $B \to X_c l \nu_l$ decays can be used in predictions of kinematic quantities in $B \to X_u l \nu_l$ decays. There are nine additional independent matrix elements at $\mathcal{O}(1/m_b^4)$ [13] and 18 at $\mathcal{O}(1/m_b^5)$ [14].

To obtain the $W_i$, the imaginary part of the $T_i$ has to be computed. This imaginary part stems from the $i\epsilon$ in the charm quark propagator and thus, the imaginary part of powers of the charm quark propagator need to be considered:

$$-\frac{i}{\pi}\text{Im}\left(\frac{1}{\Delta_0}\right)^{n+1} = \frac{(-1)^n}{n!}\delta^{(n)}\left((m_b v - q)^2 - m_c^2\right). \tag{28}$$

Here, $\delta^{(n)}$ denotes the $n$th derivative of the Dirac delta distribution.

Combinig Eqs. (15), (22) and (28) the triple differential decay rate can be expressed as

$$\frac{d^3\Gamma}{dz \, dq^2 \, dv \cdot q} = \sum_{n=0} \frac{d^3\Gamma^{(n)}}{dz \, dq^2 \, dv \cdot q}\delta^{(n)}\left((m_b v - q)^2 - m_c^2\right). \tag{29}$$

By combining Eq. (29) with analytic expressions for the $T_i$ from Ref. [13], the total semileptonic decay rate, the forward-backward asymmetry, as well as moments of distributions can

be computed. To this end, the four-momentum transfer $q$ is expressed through $Q$ and the observable of interest needs to be expressed in terms of $z$, $v \cdot Q$ and $Q^2$, leading to

$$\frac{d\langle \mathcal{O}\rangle}{dz} = \sum_{n=0} \int dQ^2 \int dv \cdot Q\, \mathcal{O}\left(z, Q^2, v \cdot Q\right) \frac{d^3\Gamma^{(n)}}{dz\, dQ^2\, dv \cdot Q} \delta^{(n)}\left(Q^2 - m_c^2\right). \tag{30}$$

The lower and upper boundaries of the $v \cdot Q$ integration are given by $\sqrt{Q^2}$ and $(m_b^2 + Q^2)/(2m_b)$, respectively. Thus, the integration over $v \cdot Q$ is performed first, followed by the integration over $Q^2$, employing the relation

$$\int dx\, \delta^{(n)}(x - x_0) f(x) = (-1)^n \frac{d^n f(x)}{dx^n}\bigg|_{x=x_0}. \tag{31}$$

## 2.3 The decay width and forward-backward asymmetry

First the total semileptonic decay rate and $\mathcal{A}_{FB}$ are computed, for which $\mathcal{O}\left(z, Q^2, v \cdot Q\right) = 1$. Performing all integrations leads to

$$\begin{aligned}
\Gamma = \frac{G_F^2 |V_{cb}|^2 m_b^5}{192\pi^3}\bigg[ &\left(1 - 8\rho - 12\rho^2 \ln\rho + 8\rho^3 - \rho^4\right)\left(1 - \frac{\hat{\mu}_\pi^2}{2m_b^2}\right) \\
&+ \frac{\hat{\mu}_G^2}{2m_b^2}\left(-3 + 8\rho - \rho^2(24 + 12\ln\rho) + 24\rho^3 - 5\rho^4\right) \\
&+ \frac{\hat{\rho}_D^3}{6m_b^3}\left(77 + 48\ln\rho - 88\rho + \rho^2(24 + 36\ln\rho) - 8\rho^3 - 5\rho^4\right) \\
&+ \frac{\hat{\rho}_{LS}^3}{2m_b^3}\left(3 - 8\rho + \rho^2(24 + 12\ln\rho) - 24\rho^3 + 5\rho^4\right) + \mathcal{O}\left(1/m_b^4\right)\bigg],
\end{aligned} \tag{32}$$

$$\begin{aligned}
\mathcal{A}_{FB} = \frac{1}{4}&\left(1 - 12\rho + 64\rho^{3/2} + \rho^2(-186 + 12\ln\rho)\right) \\
&+ \frac{\hat{\mu}_\pi^2}{3m_b^2}\left(-4 + 24\sqrt{\rho} - 92\rho + 272\rho^{3/2} - \rho^2(796 + 48\ln\rho)\right) \\
&+ \frac{\hat{\mu}_G^2}{3m_b^2}\left(-7 + 24\sqrt{\rho} - 80\rho + 224\rho^{3/2} - \rho^2(763 + 66\ln\rho)\right) \\
&+ \frac{\hat{\rho}_D^3}{3m_b^3}\Big(-14 - 6\ln\rho + 16\sqrt{\rho} - \rho(3 - 24\ln\rho) - \rho^{3/2}(488 + 384\ln\rho) + \rho^2(1640 \\
&+ 1020\ln\rho - 144\ln^2\rho)\Big) + \frac{\hat{\rho}_{LS}^3}{m_b^3}\left(-1 - 24\rho^{3/2} + \rho^2(51 - 18\ln\rho)\right) + \mathcal{O}\left(1/m_b^4, \rho^{5/2}\right),
\end{aligned} \tag{33}$$

where $\rho = m_c^2/m_b^2$ and the factor of $1/\Gamma$ in Eq. (1) has been expanded in $1/m_b$ and in $\rho$.[3] Starting from $\mathcal{O}(1/m_b^3)$, the limit $\rho \to 0$ is not finite and thus, these expressions can not be directly applied to $b \to u$ transitions. In this case weak annihilation contributions need to be included [38].

As noted in Ref. [33], $\hat{\mu}_\pi^2$ and $\hat{\mu}_G^2$ enter the decay width and the forward-backward asymmetry with the same sign and coefficients of similar magnitude, after inserting physical quark masses ($\rho \approx 0.047$).

---

[3] Although the term proportional to $\hat{\mu}_G^2$ in Eq. (33) disagrees with Eq. (3.3) in Ref. [33], it agrees with the expressions provided in the Appendix of Ref. [33].

Table 1: Numerical values for the quark masses and non-perturbative parameters, taken from Ref. [27].

| Parameter | $m_b$ (GeV) | $m_c$ (GeV) | $\hat{\mu}_\pi^2$ (GeV$^2$) | $\hat{\mu}_G^2$ (GeV$^2$) | $\hat{\rho}_D^3$ (GeV$^3$) | $\hat{\rho}_{LS}^3$ (GeV$^3$) |
|---|---|---|---|---|---|---|
| Value | 4.573 | 1.092 | 0.477 | 0.306 | 0.185 | -0.130 |
| Uncertainty | 0.012 | 0.008 | 0.056 | 0.050 | 0.031 | 0.092 |

Using the most recent central values for the non-perturbative parameters from Ref. [27], summarized in Table 1, they are predicted in the SM to be:

$$\Gamma \approx \frac{G_F^2 \, |V_{cb}|^2 \, m_b^5}{192\pi^3} \cdot 0.657 \Big( 1 - 0.011|_{\hat{\mu}_\pi^2} - 0.028|_{\hat{\mu}_G^2} - 0.032|_{\hat{\rho}_D^3} - 0.003|_{\hat{\rho}_{LS}^3} + \mathcal{O}\big(1/m_b^4\big) \Big), \quad (34)$$

$$\mathcal{A}_{FB} \approx 0.184 \Big( 1 - 0.049|_{\hat{\mu}_\pi^2} - 0.102|_{\hat{\mu}_G^2} + 0.024|_{\hat{\rho}_D^3} + 0.008|_{\hat{\rho}_{LS}^3} + \mathcal{O}\big(1/m_b^4\big) \Big), \quad (35)$$

where, the subscripts indicate the non-perturbative parameter leading to each contribution. Compared to the total rate, $\mathcal{A}_{FB}$ receives larger contributions from the $\mathcal{O}(1/m_b^2)$ corrections but smaller corrections from the Darwin term. The $\hat{\mu}_G^2$ and $\hat{\mu}_\pi^2$ dependence of $\Gamma$ and $\mathcal{A}_{FB}$ differs from that of the hadronic mass moments or moments of the electron energy spectrum (see Sec. 2.4). Because of this, measurements of $\mathcal{A}_{FB}$ and other observables sensitive to the non-perturbative parameters $\hat{\mu}_\pi^2$, $\hat{\mu}_G^2$ and $\hat{\rho}_D^3$ – which are known only at the 10% and 20% level – can improve future global fits.

## 2.4 Sums and differences of partial moments

To compute the sums and differences of partial moments $\langle \mathcal{O}^n \rangle_\pm$ defined in Eq. (3) using the phase space parametrization in Eq. (30), the observables of interest must be expressed in terms of $z$, $Q^2$ and $v \cdot Q$. The four-momentum transfer square, the hadronic mass, and the lepton energy are given in terms of these quantities by

$$q^2\big(z, Q^2, v \cdot Q\big) = Q^2 - 2m_b v \cdot Q + m_b^2, \quad (36)$$

$$M_X^2\big(z, Q^2, v \cdot Q\big) = (M_B - m_b)^2 + 2(M_B - m_b)v \cdot Q + Q^2, \quad (37)$$

$$E_l\big(z, Q^2, v \cdot Q\big) = \frac{1}{2}\Big(m_b - v \cdot Q - z\sqrt{(v \cdot Q)^2 - Q^2}\Big), \quad (38)$$

respectively.

The analytic expressions for the sums $\langle \mathcal{O}^n \rangle_+$ are lengthy and can be found in Refs. [29,33]. Two of the sums, $\langle E_l \rangle_+$ and $\langle M_X^2 \rangle_+$, as well as higher moments of the respective distributions are included in current global HQE fits [23–27]. A preliminary determination of $|V_{cb}|$ based on $\langle q^2 \rangle_+$ and higher moments of the $q^2$-spectrum has been presented in Ref. [32]. Factoring out the $m_b$-dependence of the leading order in the HQE and inserting physical values for the quark masses in the power suppressed terms leads to

$$\langle q^2 \rangle_+ \approx 0.224 m_b^2 \Big( 1 - 0.037|_{\hat{\mu}_G^2} - 0.042|_{\hat{\rho}_D^3} - 0.003|_{\hat{\rho}_{LS}^3} + \mathcal{O}\big(1/m_b^4\big) \Big), \quad (39)$$

$$\langle E_l \rangle_+ \approx 0.310 m_b \Big( 1 + 0.011|_{\hat{\mu}_\pi^2} - 0.017|_{\hat{\mu}_G^2} - 0.012|_{\hat{\rho}_D^3} - 0.0004|_{\hat{\rho}_{LS}^3} + \mathcal{O}\big(1/m_b^4\big) \Big), \quad (40)$$

$$\langle M_X^2 \rangle_+ \approx 0.210 m_b^2 \Big( 1 - 0.073|_{\hat{\mu}_\pi^2} + 0.031|_{\hat{\mu}_G^2} + 0.040|_{\hat{\rho}_D^3} + 0.003|_{\hat{\rho}_{LS}^3} + \mathcal{O}\big(1/m_b^4\big) \Big), \quad (41)$$

for the regular moments. Here $M_B \approx 5.279\,\text{GeV} \approx 1.16 m_b$ was used for the numerical estimate. The HQE parameters $\hat{\mu}_\pi^2$ and $\hat{\mu}_G^2$ enter $\langle E_l \rangle_+$ and $\langle M_X^2 \rangle_+$ with opposite signs, whereas

they enter $\mathcal{A}_{FB}$ with the same sign. Further, the dependence on $\hat{\rho}_D^3$ is stronger for all three $\langle \mathcal{O} \rangle_+$ than for $\mathcal{A}_{FB}$, while their dependence on $\hat{\mu}_G^2$ is weaker than for $\mathcal{A}_{FB}$. As a consequence, adding $\mathcal{A}_{FB}$ to a global fit has the potential to improve the determination of $\hat{\mu}_G^2$.

The analytic expressions for the differences $\langle \mathcal{O}^n \rangle_-$ presented for the first time in Appendix A. Factoring out the $m_b$-dependence of the leading order in the HQE and substituting physical values for the quark masses in the power suppressed terms gibes

$$\left\langle q^2 \right\rangle_- \approx 0.051 m_b^2 \left(1 - 0.077|_{\hat{\mu}_\pi^2} - 0.158|_{\hat{\mu}_G^2} + 0.014|_{\hat{\rho}_D^3} + 0.016|_{\hat{\rho}_{LS}^3} + \mathcal{O}\left(1/m_b^4\right)\right), \quad (42)$$

$$\left\langle E_l \right\rangle_- \approx 0.128 m_b \left(1 + 0.004|_{\hat{\mu}_\pi^2} - 0.061|_{\hat{\mu}_G^2} + 0.004|_{\hat{\rho}_D^3} + 0.005|_{\hat{\rho}_{LS}^3} + \mathcal{O}\left(1/m_b^4\right)\right), \quad (43)$$

$$\left\langle M_X^2 \right\rangle_- \approx 0.037 m_b^2 \left(1 - 0.123|_{\hat{\mu}_\pi^2} - 0.008|_{\hat{\mu}_G^2} + 0.053|_{\hat{\rho}_D^3} - 0.004|_{\hat{\rho}_{LS}^3} + \mathcal{O}\left(1/m_b^4\right)\right). \quad (44)$$

Because the difference $\left\langle q^2 \right\rangle_-$, unlike $\left\langle q^2 \right\rangle_+$, is not RPI, it depends on the parameter $\hat{\mu}_\pi^2$. As in the case of $\mathcal{A}_{FB}$, $\left\langle q^2 \right\rangle_-$ depends on the sum of $\hat{\mu}_\pi^2$ and $\hat{\mu}_G^2$ and the ratio of their coefficients in the two observables is comparable in magnitude. Further, it depends less strongly on $\hat{\rho}_D^3$ than $\mathcal{A}_{FB}$ but receives the largest relative contribution from $\hat{\rho}_{LS}^3$ of all the moments. $\left\langle E_l \right\rangle_-$ is nearly independent of $\hat{\mu}_\pi^2$, as well as the two parameters entering at $\mathcal{O}(1/m_b^3)$ and thus might provide additional information on $\hat{\mu}_G^2$. Finally, $\left\langle M_X^2 \right\rangle_-$ is sensitive to $\hat{\mu}_\pi^2$ and $\hat{\rho}_D^3$.

## 2.5 Moments of the angular distribution

Aside from the observables discussed in section 2.4, it is possible to construct moments of the angular distribution. The $\langle z^n \rangle_-$ for $n$ even and the $\langle z^n \rangle_+$ for $n$ odd are proportional to $W_3$ (and hence $\mathcal{A}_{FB}$). The other moments of $z$, however, are different linear combinations of $W_1$ and $W_2$ and Ref. [33] argues that they cannot provide any additional information beyond what is already contained in the hadronic mass moments or lepton energy moments. However, as can be seen from Eqs. (36) - (38), these observables introduce additional factors of $Q^2$ and $v \cdot Q$, which is not the case for the $\langle z^n \rangle_\pm$. In principle, one of these moments could provide additional information on the HQE parameters, while all others would be determined by the total rate and the moment of choice.

Because measurement uncertainties grow for higher moments, it is advantageous to use the lowest moments possible, which are:

$$\langle z \rangle_- \approx -0.438 \left(1 + 0.006|_{\hat{\mu}_\pi^2} - 0.009|_{\hat{\mu}_G^2} - 0.004|_{\hat{\rho}_D^3} - 0.001|_{\hat{\rho}_{LS}^3} + \mathcal{O}\left(1/m_b^4\right)\right), \quad (45)$$

$$\langle z^2 \rangle_+ \approx 0.267 \left(1 + 0.010|_{\hat{\mu}_\pi^2} - 0.015|_{\hat{\mu}_G^2} - 0.007|_{\hat{\rho}_D^3} - 0.002|_{\hat{\rho}_{LS}^3} + \mathcal{O}\left(1/m_b^4\right)\right). \quad (46)$$

Unfortunately, both of these moments are dominated by the leading-order HQE contribution, and thus do not provide additional information on the HQE parameters. They may, however, still provide constraints on interactions beyond the standard left-handed current. Such a study is beyond the scope of this work.

# 3 Implications of an experimental cut on the momentum transfer

In the previous section the dependence of $\mathcal{A}_{FB}$ and the various moments on the non-perturbative parameters was discussed in the idealistic scenario in which no cut on the leptonic phase-space is applied. Typically, experiments apply a cut on the lepton energy when measuring moments of the lepton-energy spectrum or the hadronic invariant mass moments [15,17–22], introducing a non-trivial dependence of the angular spectrum on the chosen minimum energy. This leads to a discontinuity in the angular spectrum that depends upon the

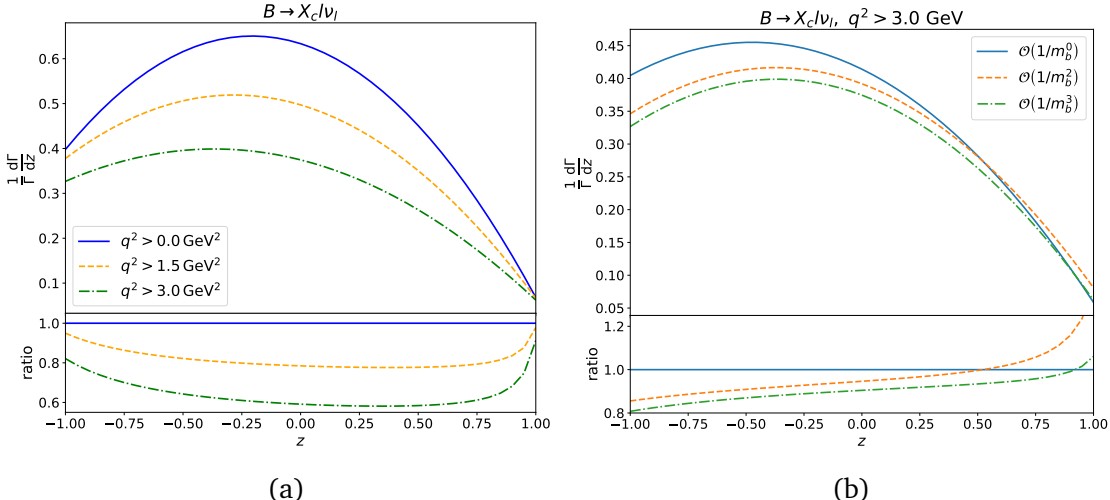

Figure 2: Left: differential decay rate for $B \to X_c l\, \nu_l$ at $\mathcal{O}(1/m_b^3)$ for different $q^2$ cuts. Right: the differential decay rate for $B \to X_c l\, \nu_l$ at different orders in the HQE for $q_{\text{cut}}^2 = 3\ \text{GeV}^2$. In both plots, the lower panel shows the ratios of the dashed curves to the solid one.

chosen minimum energy [33], which can lead to problems when unfolding data or for finding agreement between measured and simulated spectra. Further, the lepton-energy cut introduces an unphysical asymmetry in the angular spectrum, which could lead to difficulties in extracting $\mathcal{A}_{FB}$ from a measurement.

As an alternative to the minimum lepton energy cut, a cut on $q^2$ was suggested in [29] and implemented in the measurement of $q^2$ moments with Belle [31]. Although the cut on $q^2$ was motivated by a desire to preserve the RPI of the moments $\left\langle \left(q^2\right)^n \right\rangle_+$, it also has advantages for a measurement of $\mathcal{A}_{FB}$, addressing the aforementioned problems associated with a minimum lepton energy cut, as shown below.

## 3.1 Differential decay rate

A cut on $q^2$ enters the calculation of the observables in Sec. 2 through the upper integration boundary of the $v \cdot Q$ integration in Eq. (30). The new integration boundary is given by $(m_b^2 + Q^2 - q_{\text{cut}}^2)/(2m_b)$. This simplifies the $v \cdot Q$ and $z$ integration in comparison to the case of an $E_l$ cut as only one integration region exists instead of three [33]. Further, the resulting spectrum is a quadratic polynomial in $z$ as in the case without a cut and will be refered to as "angular" spectrum in the following.

In the left panel of Fig. 2 the angular spectrum is shown for different choices of $q_{\text{cut}}^2$. Even with a cut on $q^2$, the spectrum is smooth and well-behaved. Although the available phase-space decreases with increasing $q_{\text{cut}}^2$, the shape of the spectrum stays qualitatively the same. The peak of the spectrum also becomes less pronounced and shifts in the negative $z$ direction.

The right panel of Fig. 2 shows the contributions of different orders in the HQE for the case of $q_{\text{cut}}^2 = 3\ \text{GeV}^2$. The $\mathcal{O}\left(1/m_b^2\right)$ corrections are sizeable, especially for $|z| \to 1$.

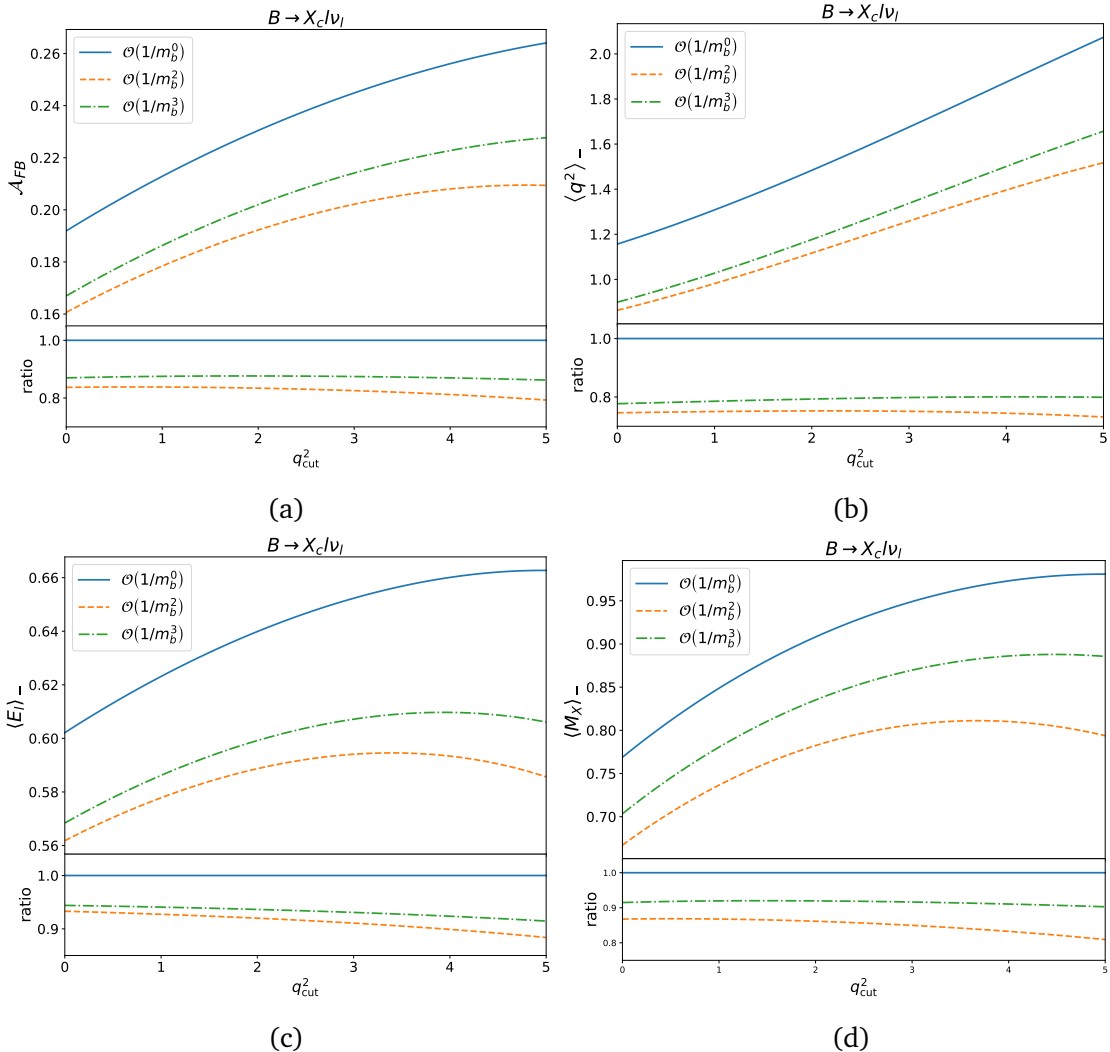

Figure 3: $\mathcal{A}_{FB}$ and the $\langle\mathcal{O}\rangle_-$ as a function of $q^2_{\text{cut}}$. The solid, dashed and dash-dotted lines include contributions through $\mathcal{O}(1/m_b^0)$, $\mathcal{O}(1/m_b^2)$ and $\mathcal{O}(1/m_b^3)$, respectively. The lower panels show each curve normalized to the respective $\mathcal{O}(1/m_b^0)$ curve.

## 3.2 Numerical results for $\mathcal{A}_{FB}$ and the moments

For the measurement of the $\langle q^2\rangle_+$ moments at Belle, the lowest cut considered was $q^2_{\text{cut}} = 3$ GeV$^2$ [31]. Although it is possible to decrease this cut, experimental and modelling uncertainties grow rapidly as $q^2_{\text{cut}}$ is further reduced [39]. Additionaly, lepton identification gets worse for lower values of $q^2_{\text{cut}}$, as discussed in section 4.2. Consequently, numerical results in this section are presented for $q^2_{\text{cut}} = 3$ GeV$^2$, as well as for $q^2_{\text{cut}} = 5$ GeV$^2$.

In Fig. 3 the $q^2_{\text{cut}}$ dependence of $\mathcal{A}_{FB}$ and the $\langle\mathcal{O}\rangle_-$ is shown. For all four observables, the $\mathcal{O}(1/m_b^2)$ corrections are negative and grow in magnitude for increasing $q^2_{\text{cut}}$. This growth is compensated, however, by a growth of the positive $\mathcal{O}(1/m_b^3)$ corrections. Consequently, the sum of higher-order contributions in the HQE is nearly independent of $q^2_{\text{cut}}$ in the range under consideration.

The sizes of corrections stemming from the individual non-perturbative parameters to $\mathcal{A}_{FB}$ and the $\langle\mathcal{O}\rangle_-$ are summarized for three different values of $q^2_{\text{cut}}$ in Table 2. For all four observables, the relative contributions remain stable for increasing $q^2_{\text{cut}}$. Consequently, for the

extraction of the non-perturbative parameters the exact choice of $q_{\text{cut}}^2$ only has a mild impact and can be tuned to minimize experimental uncertainties.

Table 2: $\mathcal{O}\left(1/m_b^3\right)$ contributions to $\mathcal{A}_{FB}$ and the differences $\langle \mathcal{O} \rangle_-$ from non-perturbative parameters. The contributions are given as percentage of the $\mathcal{O}\left(1/m_b^0\right)$ results.

| $q_{\text{cut}}^2$ (GeV$^2$) | $\mathcal{A}_{FB}$ | | | $\langle q^2 \rangle_-$ | | |
|---|---|---|---|---|---|---|
| | 0 | 3 | 5 | 0 | 3 | 5 |
| $\hat{\mu}_\pi^2$ | −4.9% | −5.9% | −7.7% | −7.7% | −8.1% | −9.6% |
| $\hat{\mu}_{\text{G}}^2$ | −10.2% | −10.5% | −12.0% | −15.8% | −14.9% | −15.6% |
| $\hat{\rho}_{\text{D}}^3$ | +2.4% | +3.8% | +5.4% | +1.4% | +2.8% | +4.4% |
| $\hat{\rho}_{\text{LS}}^3$ | +0.8% | +1.1% | +1.6% | +1.6% | +1.8% | +2.3% |
| $q_{\text{cut}}^2$ (GeV$^2$) | $\langle E_l \rangle_-$ | | | $\langle M_X^2 \rangle_-$ | | |
| | 0 | 3 | 5 | 0 | 3 | 5 |
| $\hat{\mu}_\pi^2$ | +0.1% | −0.01% | −0.9% | −12.3% | −13.8% | −16.3% |
| $\hat{\mu}_{\text{G}}^2$ | −6.1% | −7.7% | −9.5% | −0.1% | −1.1% | −2.7% |
| $\hat{\rho}_{\text{D}}^3$ | +0.4% | +1.0% | +1.6% | +5.3% | +7.3% | +10.1% |
| $\hat{\rho}_{\text{LS}}^3$ | +0.5% | +0.8% | +1.2% | −0.4% | −0.3% | −0.1% |

For $\mathcal{A}_{FB}$, the $\mathcal{O}(1/m_b^2)$ corrections reduce the value of $\mathcal{A}_{FB}$ by 16% to 21%, while the $\mathcal{O}(1/m_b^3)$ increase $\mathcal{A}_{FB}$ by 3% to 7%. While for the range under consideration, $\mathcal{A}_{FB}$ is most sensitive to $\hat{\mu}_{\text{G}}^2$, the relative contribution decreases from 57% of the total corrections from higher orders in the HQE at $q_{\text{cut}}^2 = 0$ GeV$^2$ to 42% at $q_{\text{cut}}^2 = 5$ GeV$^2$. At $q_{\text{cut}}^2 = 3$ GeV$^2$ the contribution from $\hat{\mu}_{\text{G}}^2$ still amounts to more than half of the power suppressed contributions, making this choice a good compromise between experimental sensitivity and systematics.

The difference of partial moments most sensitive to higher order corrections in the HQE is $\langle q^2 \rangle_-$. As shown in the upper right panel of Fig. 3, the $\mathcal{O}(1/m_b^2)$ corrections decrease $\langle q^2 \rangle_-$ by 25%, while the $\mathcal{O}(1/m_b^3)$ corrections increase it by 3%, relative to the leading order in the HQE. Like $\mathcal{A}_{FB}$, $\langle q^2 \rangle_-$ receives the largest contribution from $\hat{\mu}_{\text{G}}^2$, throughout the whole range under consideration. For $q_{\text{cut}}^2 = 0, 3, 5$ GeV$^2$ the contribution from $\hat{\mu}_{\text{G}}^2$ comprises 60%, 55% and 50% of the higher order corrections, respectively. The difference $\langle q^2 \rangle_-$ is also sensitive to the spin-orbit coupling $\hat{\rho}_{\text{LS}}^3$, for $q_{\text{cut}}^2 = 0, 3, 5$ GeV$^2$ it contributes to 6.0%, 6.7% and 7.4% of the higher order corrections, respectively.

Of the four observables shown in Fig. 3, $\langle E_l \rangle_-$ receives the smallest contributions from higher-order terms in the HQE. Through $\mathcal{O}(1/m_b^3)$ the total corrections range between 6% and 10%. Without a $q^2$ cut, 80% of these corrections are due to $\hat{\mu}_{\text{G}}^2$ and 6.3% are due to $\hat{\rho}_{\text{LS}}^3$. These fractions increase roughly linearly with $q_{\text{cut}}^2$. Hence, similar to $\langle q^2 \rangle_-$ and $\mathcal{A}_{FB}$ a precise measurement of $\langle E_l \rangle_-$ can provide information on the less well known non-perturbative parameters.

Lastly, the higher-order HQE corrections to $\langle M_X^2 \rangle_-$ are dominated by $\hat{\mu}_\pi^2$ and $\hat{\rho}_{\text{D}}^3$. Therefore, $\langle M_X^2 \rangle_-$ provides similar information to the moments already employed in global fits.

For all three observables considered, the differences $\langle \mathcal{O} \rangle_-$ receive larger contributions from the higher orders in the HQE than the sums $\langle \mathcal{O} \rangle_+$. Thus, even a moderately precise measurement could aid the global fit to determine the non-perturbative HQE parameters. With the current uncertainties of 10% to 20% for the non-perturbative parameters, measurements of

$\mathcal{A}_{FB}$ and $\left\langle q^2 \right\rangle_-$ would need to reach a level of precision of 3% to improve the global fit. For comparison, the uncertainty on $\left\langle q^2 \right\rangle_+$ currently is smaller than 2% for $q^2_{\text{cut}} > 3\,\text{GeV}^2$ and smaller than 1% for $q^2_{\text{cut}} > 6\,\text{GeV}^2$ [31].

### 3.3 Subtraction of the $X_u$ component from $B \to X l \nu_l$

For most experimental determinations of kinematic distributions in inclusive $B \to X_c l \nu_l$ decays, the distribution is infact measured in $B \to X l \nu_l$ decays, where $X \equiv X_c + X_u$. The $X_u$ component is estimated via Monte Carlo simulations and subtracted from the data. This procedure introduces a modelling uncertainty that is non-negligble [31].

Recently, it was suggested in Ref. [40] to instead measure observables in $B \to X l \nu_l$ decays and subtract the $B \to X_u l \nu_l$ component using HQE based predictions . Using the linearity of the numerator and denominator in Eq. (3) with respect to the $X_c$ and $X_u$ components, $\mathcal{A}_{FB}$ and any sum or difference of partial moments measured in $B \to X l \nu_l$ decays can be written as

$$\mathcal{A}_{FB} = \frac{\Gamma_c \mathcal{A}^c_{FB} + \Gamma_u \mathcal{A}^u_{FB}}{\Gamma_c + \Gamma_u}, \tag{47}$$

$$\left\langle \mathcal{O}_{\text{tot}} \right\rangle_\pm = \frac{\Gamma_c \left\langle \mathcal{O}_c \right\rangle_\pm + \Gamma_u \left\langle \mathcal{O}_u \right\rangle_\pm}{\Gamma_c + \Gamma_u}. \tag{48}$$

Solving this equation for the charm component then yields

$$\mathcal{A}^c_{FB} = \mathcal{A}^{\text{tot}}_{FB} + \frac{\Gamma_u}{\Gamma_c} \left( \mathcal{A}^{\text{tot}}_{FB} - \mathcal{A}^u_{FB} \right) = \mathcal{A}^{\text{tot}}_{FB} + R_{\text{PS}} \frac{|V_{ub}|^2}{|V_{cb}|^2} \left( \mathcal{A}^{\text{tot}}_{FB} - \mathcal{A}^u_{FB} \right), \tag{49}$$

$$\left\langle \mathcal{O}_c \right\rangle_\pm = \left\langle \mathcal{O}_{\text{tot}} \right\rangle_\pm + \frac{\Gamma_u}{\Gamma_c} \left( \left\langle \mathcal{O}_{\text{tot}} \right\rangle_\pm - \left\langle \mathcal{O}_u \right\rangle_\pm \right) = \left\langle \mathcal{O}_{\text{tot}} \right\rangle_\pm + R_{\text{PS}} \frac{|V_{ub}|^2}{|V_{cb}|^2} \left( \left\langle \mathcal{O}_{\text{tot}} \right\rangle_\pm - \left\langle \mathcal{O}_u \right\rangle_\pm \right). \tag{50}$$

The ratio $|V_{ub}|^2/|V_{cb}|^2$ is of the order of one percent, while $R_{\text{PS}}$ at $\mathcal{O}(1/m_b^2)$ in the HQE ranges from 1.42 at $q^2_{\text{cut}} = 0\,\text{GeV}^2$ to 1.76 at $q^2_{\text{cut}} = 5\,\text{GeV}^2$. For $\mathcal{A}_{FB}$, the second term in Eq. (49) is further suppressed, since $\mathcal{A}^{\text{tot}}_{FB} - \mathcal{A}^u_{FB} \approx -0.1 \mathcal{A}^{\text{tot}}_{FB}$ for $q^2_{\text{cut}} = 3\,\text{GeV}^2$. Similarly, for observables with $\left\langle \mathcal{O}_u \right\rangle_\pm \approx \left\langle \mathcal{O}_{\text{tot}} \right\rangle_\pm$, the second term in Eq. (50) is further suppressed. This is the case for the other three observables. In total, the second term in Eqs. (49) and (50) amounts to a sub-percent level shift.

Once a measurement of $\mathcal{A}_{FB}$ in $B \to X l \nu_l$ decays becomes available and reaches sub-percent level precision, the prediction of the $b \to u l \nu_l$ component can be extended to $\mathcal{O}(1/m_b^3)$ by including weak annihilation contributions, following Ref. [38].

## 4 Towards a measurement of $\mathcal{A}_{FB}$ at Belle II

The previous sections discussed the forward-backward asymmetry and differences of partial moments on a fully inclusive level. Before comparisons with experiment can be made, however, effects such as final-state radiation, background processes and detector acceptance effects must be taken into account. Here, two issues relevant for connecting measurements of $\mathcal{A}_{FB}$ and the $\left\langle \mathcal{O} \right\rangle_-$ to predictions of the HQE are discussed: the impact of final-state radiation and particle-identification requirements. The relevant background processes and impact of detector acceptance depend strongly on the analysis details, and are therefore beyond the scope of this work.

To extract the angular distribution or $\mathcal{A}_{FB}$ from measurements at Belle II, Monte Carlo simulations of signal and background events including final-state radiation and detector simulations are required. In this section the event generator Sherpa [41, 42] is used to simulate

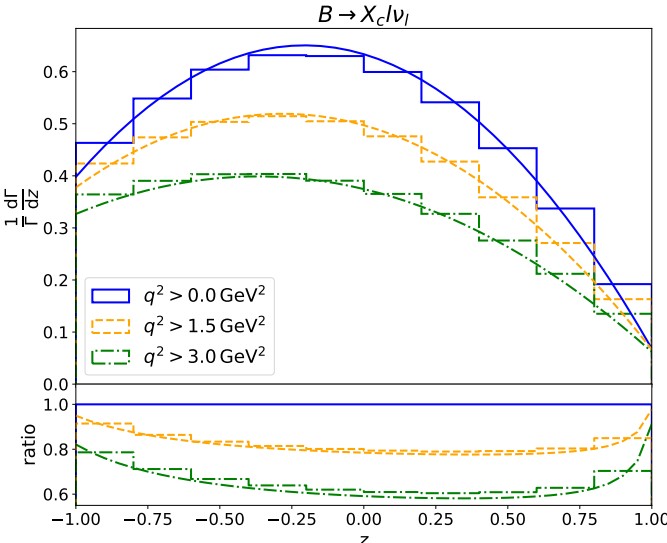

Figure 4: The differential decay rate for three values of $q^2_{\text{cut}}$ normalized to the total decay rate ($q^2_{\text{cut}} = 0$ GeV$^2$). The solid and dashed curves are the same HQE predictions shown in Fig. 2, while the histograms are the simulation results from `Sherpa`. The lower panel shows the ratios of the dashed curves to the solid one.

the production of the $\Upsilon(4S)$ resonance in asymmetric electron-positron collisions, the subsequent $\Upsilon(4S)$-decay into pairs of neutral or charged $B$-mesons, and the $B$-mesons' respective semileptonic decays. Although the Belle II experiment uses `EVTGEN` [43] to simulate decays of $B$ mesons and `PHOTOS` [44, 45] to treat final-state radiation (FSR), `Sherpa` is capable of a more advanced treatment of FSR, which is relevant for the discussion in Sec. 4.1. Particle-identification (PID) requirements are implemented in Sec. 4.2 in a simplified manner by imposing cuts on the lepton momenta.

Before considering the impact of FSR and PID, it is useful to compare the HQE predictions to the simulation data with just the cut on $q^2$. In Fig. 4 the simulated angular spectrum for $q^2_{\text{cut}} = \{0, 1.5, 3.0\}$ GeV$^2$ is shown and compared with the HQE prediction (see Fig. 2). The shapes of the distributions agrees well for all cut values.

It remains to be seen, however, whether data and simulation agree in a future measurement of the angular distribution. The angular dependence for $D$ mesons and their excited states are predicted by form factor models [46–48]. The angular spectrum of non-resonant contributions such as $B \to D^{(*)}\pi\pi l\nu_l$ decays, however, are less well known. Furthermore, there is a difference between the sum of exclusive states and the inclusive branching fraction, commonly referred to as the *gap* (see, *e.g.*, Ref. [31]), which must be taken into account. Two approaches used in experimental analyses to treat non-resonant contributions and fill the *gap* are: (1) simulating $B \to D^{(*)}\pi\pi l\nu_l$ and $B \to D^{(*)}\eta l\nu_l$ decays through the decay of two broad intermediate (yet unobserved) states [31]; (2) equally distributing all final-state particles in phase space. The two approaches not only lead to different shapes for the $q^2$, $M_X$ and $E_l$ distributions [31], but also drastically impact the angular distribution. The equidistribution of final state particles in phase space leads to a flat – hence symmetric – distribution in $z$, whereas the treatment of decays as broad intermediate states leads to a distribution similar to the one found for $B \to D^{**}l\nu_l$.

## 4.1 Impact of final-state radiation

Final-state radiation off of leptons changes the shape of the lepton-energy spectrum and, as a consequence, the angular distribution. Because electrons are two orders of magnitude lighter than muons, they radiate more photons, thereby inducing a difference between $\mathcal{A}_{FB}$ for electrons and muons. In light of the discrepancies between $\mathcal{A}_{FB}$ measured in $B \to D^* \mu \nu_\mu$ and $B \to D^* e \nu_e$ decays, such effects must be carefully analyzed. `Sherpa` implements Yennie-Frautschi-Suura resummation [49] to treat FSR and allows the inclusion of hard photons in addition to soft radiation [50]. Differences between including all FSR and including only soft radiation have been studied in the context of $B$-meson decays and compared to `PHOTOS`, showing that the soft-radiation-only mode in `Sherpa` largely agrees with `PHOTOS` [51]. Studying differences between the two modes in `Sherpa` enables an estimate of the impact of hard-photon radiation on $\mathcal{A}_{FB}$ which is not accounted for by Belle II simulations.

The effect of including FSR for $B \to X e \nu_e$ decays is shown in Fig. 5. Final-state radiation reduces the electron energy and, therefore, shifts the angular distribution towards larger $z$ values. Imposing a cut on $q^2$, however, reduces the impact of FSR, especially in the rightmost bins where the impact of hard radiation is the largest. For $q^2_{\text{cut}} = 3.0\,\text{GeV}^2$, including soft radiation reduces $\mathcal{A}_{FB}$ by 3.5%, while including both soft and hard radiation reduces $\mathcal{A}_{FB}$ by 6%.

Although these effects may seem rather large, they are mitigated in experimental analyses by combining hard photons with the electron from which they most likely originated. In Belle's recent $B \to X_c l \nu_l$ anlysis [31], the momentum of the photon with the highest energy in a $5°$ cone around an electron was added to the momentum of the electron. When the same procedure is applied to the `Sherpa` simulation data, it greatly reduces the impact of hard photons (see the lower row of Fig. 5). Without a cut on $q^2$, including hard radiation decreases $\mathcal{A}_{FB}$ by 3%; this is reduced to only 0.5% for $q^2_{\text{cut}} = 3.0\,\text{GeV}^2$. Consequently, `PHOTOS` describes FSR in $\mathcal{A}_{FB}$ adequately given the current experimental uncertainty, but FSR will need to be revisited when a measurement of $\mathcal{A}_{FB}$ with sub-percent precision becomes available.

Because photon radiation off of leptons is proportional to $\ln(m_l^2)$, muons radiate fewer photons than electrons. As shown in Fig. 6, the influence of FSR on the shape of the angular distribution for the muon decay modes is similar to that of the electron modes without clustering of collinear photons (top row of Fig. 5), but much smaller in size. As in the electron case, including a cut on $q^2$ reduces the contribution to $\mathcal{A}_{FB}$ from hard radiation from 1.5% for $q^2_{\text{cut}} = 0.0\,\text{GeV}^2$ to 0.6% for $q^2_{\text{cut}} = 3.0\,\text{GeV}^2$.

In summary, clustering electrons together with hard photons and applying a cut of $q^2 > 3.0\,\text{GeV}^2$ reduces the effect of hard radiation on $\mathcal{A}_{FB}$ to below 1%. Reducing the minimum $q^2$ increases the effect of hard photons – which are not accounted for by `PHOTOS` – and introduces a difference in $\mathcal{A}_{FB}$ between electrons and muons of around 1% to 3%.

## 4.2 Lepton identification

Another source of differences between measurements of $\mathcal{A}_{FB}$ in the electron and muon channels is the experimental particle identification. In Belle's recent $B \to X_c l \nu_l$ analysis [31] minimum transverse momenta in the laboratory frame of 300 MeV and 600 MeV were required for electrons and muons, respectively. Such a requirement cannot be included in HQE predictions, however, since only cuts on observables composed of the momenta of the $B$ meson and its decay products can be applied. This not only leads to differences between electrons and muons, but also to a mismatch between the experimental and theoretical quantities being compared. Consequently, it is crucial to correct for such requirements in experimental analyses. This could be implemented, for example, through unfolding to the underlying distribution without transverse-momentum requirement. Alternatively, cuts on kinematic quantities such

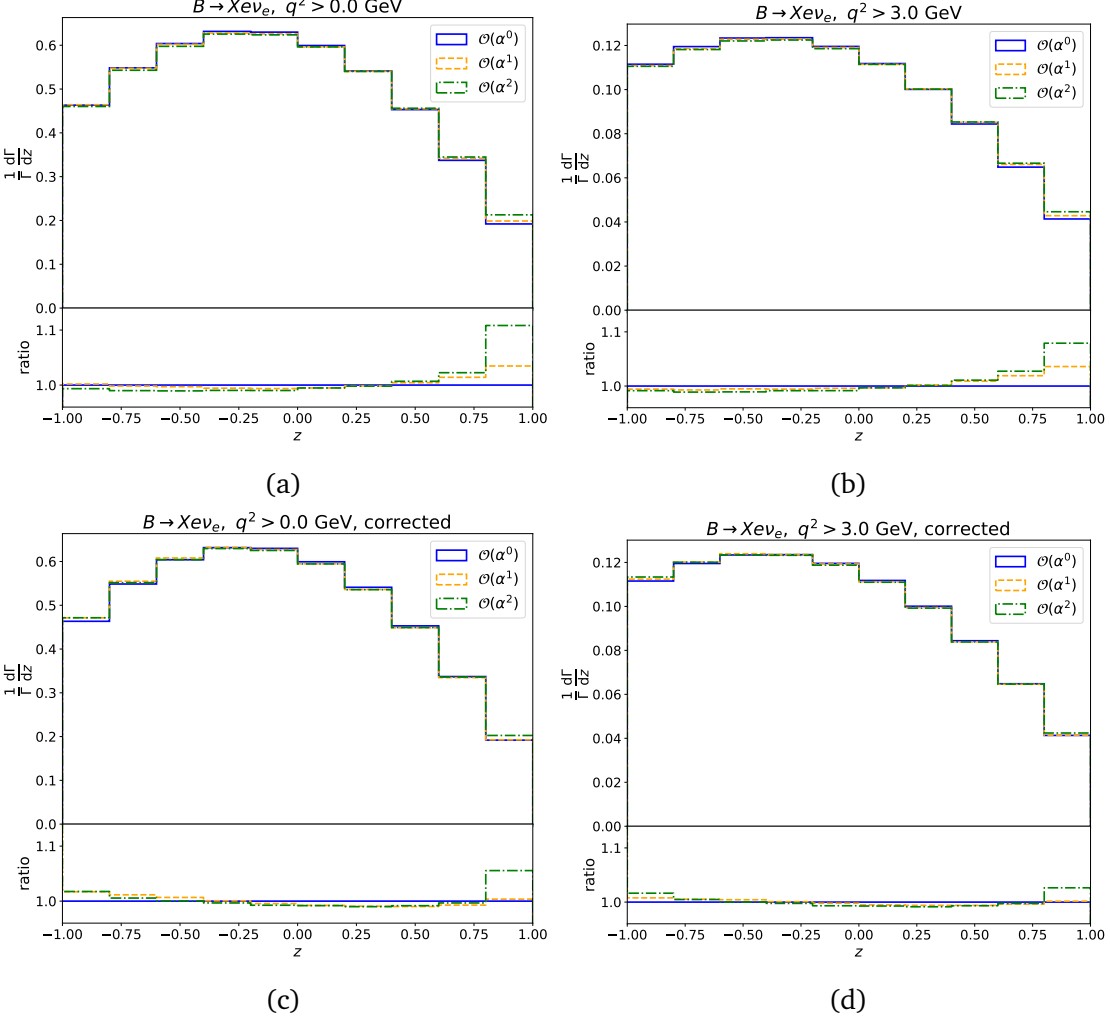

Figure 5: The differential decay rate for $B \rightarrow X e \nu_e$ decays for different treatments of the final-state radiation. The figures in the left column do not include a cut on $q^2$, while the figures in the right column include a lower cut on $q^2$ of $3\,\text{GeV}^2$. In the two figures in the upper row no correction for hard-collinear radiation was applied, whereas for the two figures in the lower row the photon with the highest energy within a $5°$ cone around the electron was combined with the electron. The labels $\mathcal{O}(\alpha^0)$, $\mathcal{O}(\alpha^1)$ and $\mathcal{O}(\alpha^2)$ denote no final-state radiation, soft final-state radiation only and a full YFS treatment, respectively.

as $E_l$ or $q^2$ can be chosen in such a way that the dependence on the transverse-momentum requirement is minimized. In the following, we choose a minimum transverse momentum of $300\,\text{MeV}$ for electrons and $500\,\text{MeV}$ for muons.[4]

The resulting simulated angular distributions for electrons are shown in the upper row of Fig. 7. The difference between distributions with and without a transverse-momentum requirement is substantial, especially for low-energy electrons ($z \approx 1$). Imposing a minimum required $q^2$ reduces the effect of the transverse momentum requirement. Even for $q^2_{\text{cut}} = 3.0\,\text{GeV}^2$, however, the difference between $\mathcal{A}_{FB}$ with and without such a $p_T$ cut is as large as 6.5%. In

---

[4]Although Belle analyses require a minimum muon transverse momentum of $600\,\text{MeV}$, muons with transverse momenta as low as $500\,\text{MeV}$ reach the $K_L$ detection system and could therefore be identified. This also holds true for Belle II.

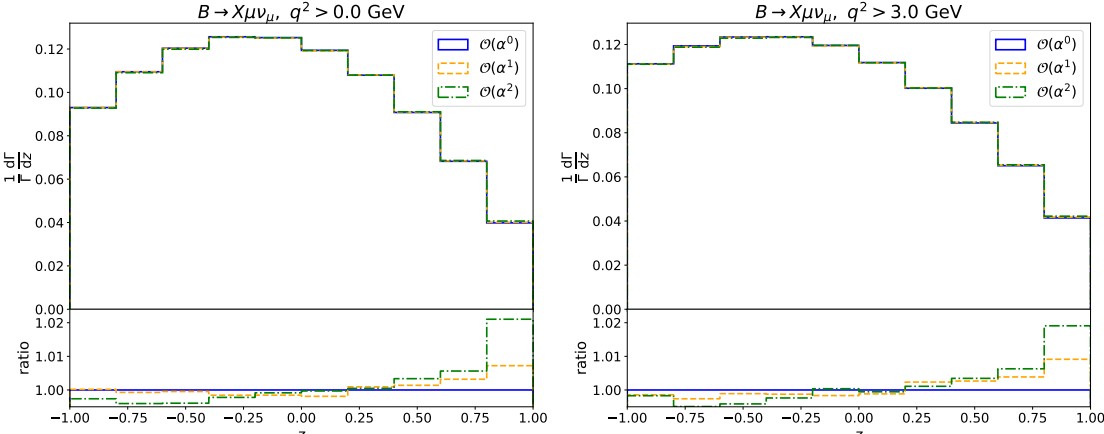

Figure 6: The differential decay rate for $B \to X\mu\nu_\mu$ decays. The left-hand figure is without a cut on $q^2$, while the right-hand figure includes a lower cut on $q^2$ of 3 GeV$^2$. The labels $\mathcal{O}(\alpha^0)$, $\mathcal{O}(\alpha^1)$ and $\mathcal{O}(\alpha^2)$ denote no final-state radiation, soft final-state radiation only and a full YFS treatment, respectively.

contrast, for observables that are not sensitive to the angle between the electron and $B$ meson, the difference is less than 0.3% for $q^2_{\text{cut}} = 3.0 \, \text{GeV}^2$.

The corresponding angular distributions for muons are shown in the lower row of Fig. 7. Here, the transverse-momentum requirement has a substantially larger impact than for electrons and leads to a difference in $\mathcal{A}_{FB}$ of 20% with $q^2_{\text{cut}} = 3.0 \, \text{GeV}^2$. For $\langle q^2 \rangle_+$, the difference is below 0.8% for $q^2_{\text{cut}} = 3.0 \, \text{GeV}^2$, but increases to 2% when $q^2_{\text{cut}} = 1.5 \, \text{GeV}^2$. The shifts in the $\langle \mathcal{O} \rangle_-$ are similar in size to those for $\mathcal{A}_{FB}$.

Recent improvements of charged lepton identification in the Belle II experiment allow to identify electrons and muons with a minimum energy in the laboratory frame of 400 MeV [52]. The lepton energy in the laboratory frame does not directly correspond to the lepton energy in the $B$-meson rest frame. Consequently, a cut on the lepton energy in the $B$-meson rest frame does not correspond to a lepton-energy requirement in the laboratory frame. However, a cut on $q^2$ can remove the dependence on the lepton-energy requirement as shown in Fig. 8.

The $q^2$ spectrum with a minimum lepton-energy requirement of 400 MeV agrees perfectly with the spectrum without any requirements for $q^2 > 3.5 \, \text{GeV}^2$. Consequently, the estimate of Ref. [29] for a minimum $q^2$ requirement to be independent of possible additional lepton-energy cuts in the $B$-meson rest frame also applies in the laboratory frame. For a lepton-energy requirement of 400 MeV the corresponding $q^2$ requirement is 3.6 GeV$^2$, whereas for a more conservative lepton-energy requirement of 500 MeV the $q^2$ requirement would be 4.5 GeV$^2$. While the lepton-energy requirement is sufficient for lepton identification, an experimental analysis still needs to apply a transverse-momentum requirement of 100 MeV to ensure that lepton tracks and, as a consequence, their momenta are properly reconstructed [53]. Fig. 9 shows the angular distribution for $q^2 > 3.0 \, \text{GeV}^2$ with and without the lepton-energy requirement of 400 MeV and transverse-momentum requirement of 100 MeV. Both distributions nearly agree perfectly and the impact of the laboratory frame requirements on $\mathcal{A}_{FB}$ are below 1%, making this choice of cuts ideal for measurements of $\mathcal{A}_{FB}$ and the differences of partial moments.

Should the preferred choice of cuts not be feasible to implement in an analysis, alternative ways to correct for transverse-momentum requirements need to be applied. A possible approach to correct experimental data for the influence of transverse-momentum requirements

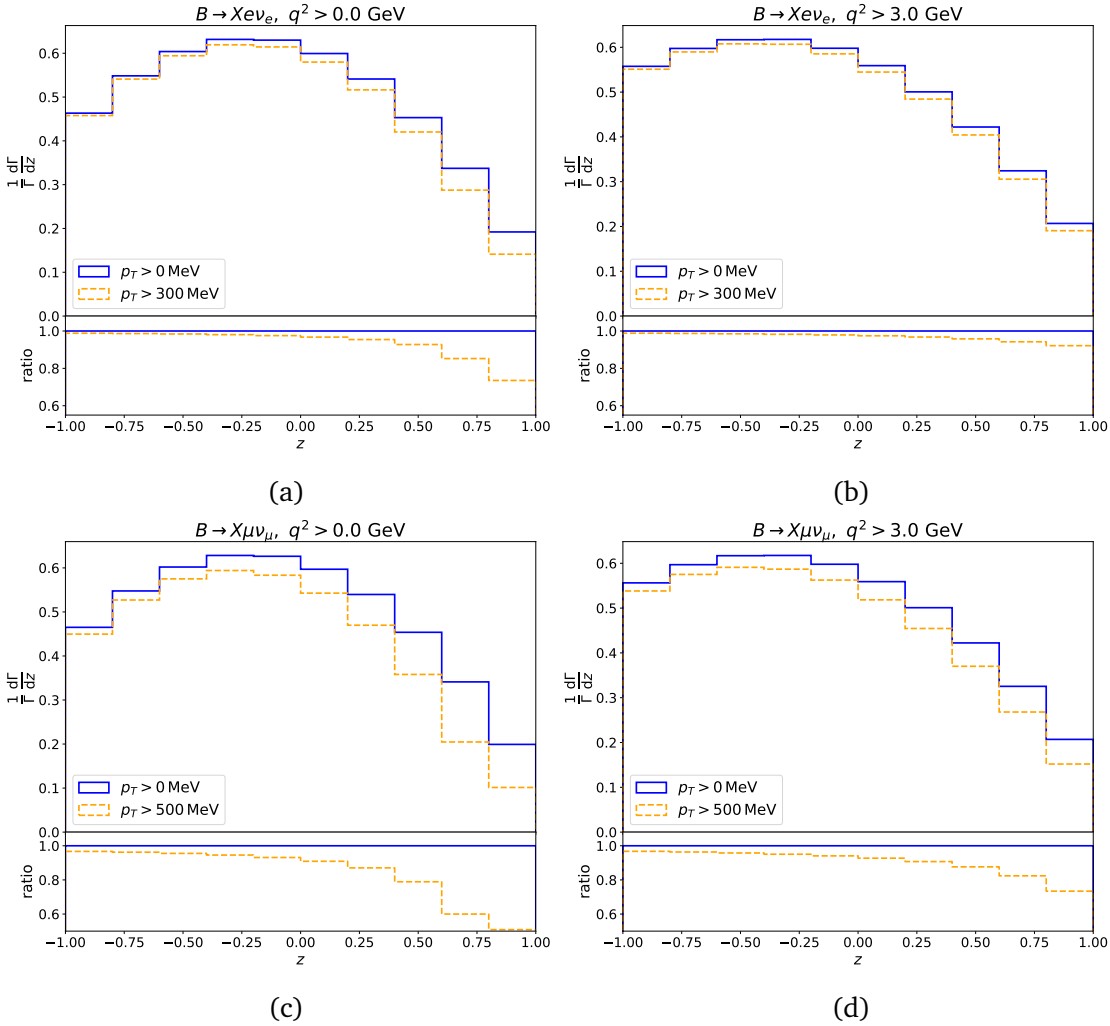

Figure 7: The differential decay rate for $B \to X l \nu_l$ decays with and without requirements on the transverse momentum of the lepton in the laboratory frame. The figures in the upper row show the rate for $B \to X e \nu_e$ for $q^2 > 0$ GeV$^2$ and $q^2 > 3$ GeV$^2$, respectively. The figures in the lower row show the rate for $B \to X \mu \nu_\mu$ with the same $q^2$ cuts as for the electron case.

(and final-state radiation) is to introduce a bias correction factor (see, *e.g.*, Refs. [19,31]) that depends on the selection cut. The correction could be obtained from simulations with and without detector effects and final-state radiation. Such a procedure might not be adequate for $\mathcal{A}_{FB}$, however, since the impact of the transverse-momentum requirement alone is a factor of 20 larger than for the $q^2$ moments and is not uniform in $z$. Alternatively, future experiments could correct measurements of the angular distribution for the transverse-momentum requirement by unfolding the distribution by means of the singular value decomposition algorithm [54] or similar methods, and then extract $\mathcal{A}_{FB}$ from the corrected distribution.

## 4.3 Comment on the tension in $\mathcal{A}_{FB}$ in $B \to D^* l \nu_l$ decays

Finally, a comment regarding the tension in $\mathcal{A}_{FB}$ measured in $B \to D^* \mu \nu_\mu$ decays is in order. In Ref. [34] the Belle collaboration provides bin-wise efficiency factors for the angular distribution of the charged lepton for both electron and muon modes. These efficiency factors correspond

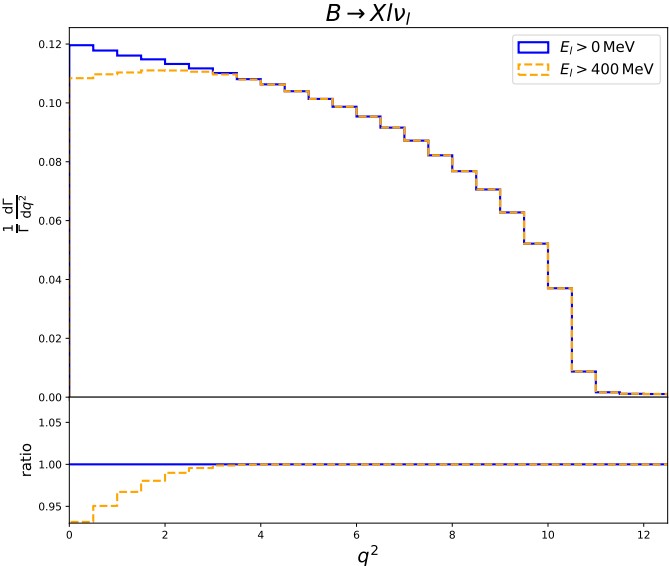

Figure 8: The $q^2$ spectrum in $B \to X l \nu_l$ decays with and without a minimum requirement on the lepton energy in the laboratory frame.

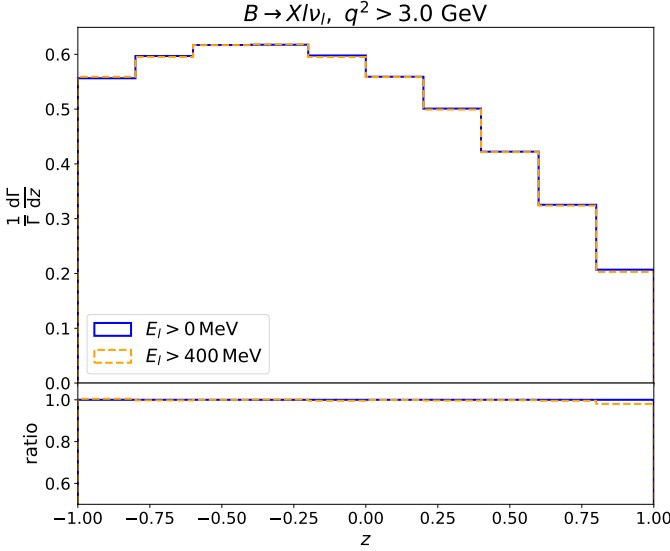

Figure 9: The angular spectrum in $B \to X l \nu_l$ decays with and without a minimum requirement on the lepton energy in the laboratory frame.

to the probability that an event passes the selection criteria and is properly reconstructed. As discussed earlier, Belle's cuts on the lepton's transverse momentum in the laboratory frame is $p_T > 300$ MeV and $p_T > 600$ MeV for electrons and muons, respectively. Figure 7 suggests that the efficiency factors for muons should be smaller than for electrons for $z > 0$.[5] Although this is true for four out of the five bins in Ref. [34], the efficiencies are almost identical in the bin closest to $z = -1$, contrary to the expectation from Fig. 7. This difference may be due to different detector effects not taken into account in the simple simulations of this section.

---

[5]When comparing the distributions in this work with those of Ref. [34], it is important to remember that $z > 0$ here corresponds to $\cos \theta_l < 0$ in that work.

It seems unlikely that the electron and muon efficiencies are almost identical, and should be further investigated, especially since a shift in the muon efficiency could account for a significant fraction of the discrepancy in $\mathcal{A}_{FB}$ for $B \to D^* \mu \nu_\mu$.

## 5 Conclusions and outlook

The analysis in this work of the lepton forward-backward asymmetry, $\mathcal{A}_{FB}$, and differences of partial moments in the forward and backward directions shows that they are interesting observables for studying semileptonic $B$-meson decays because they provide new constraints on the non-perturbative parameters of the HQE that are complementary to ones from the lepton-energy and hadronic-mass moments. Of particular note are that the parameters $\hat{\mu}_\pi^2$ and $\hat{\mu}_G^2$ enter with the same sign and that the they are numerically more sensitive to $\hat{\mu}_G^2$. Consequently, a measurement of $\mathcal{A}_{FB}$ in $B \to X_c l \nu_l$ decays will improve the inclusive determination of $|V_{cb}|$. Further, separate measurements of $\mathcal{A}_{FB}$ for electron and muon final states may shed light on the discrepancy between electron and muon modes observed in exclusive $B \to D^* l \nu_l$ decays.

This analysis also shows that a cut on the four-momentum transfer squared ($q^2$) – in contrast to the cut on lepton energy currently employed – does not introduce discontinuities in the angular distribution. Thus, the angular distribution remains a quadratic polynomial, even with the $q^2$ cut. This simplifies experimental analyses that correct for detector acceptance effects by unfolding the measured distribution to the true underlying distribution, since no discontinuities are present in the latter.

Correcting for these experimental cuts will be crucial for a future measurement of $\mathcal{A}_{FB}$ at Belle II, because lepton-momentum requirements have a significant impact on $\mathcal{A}_{FB}$ (For a cut of $q^2 > 3 \, \text{GeV}^2$, $\mathcal{A}_{FB}$ is shifted by 6% and 20% for electrons and muons, respectively). These effects can not be accounted for in the HQE and must therefore be accounted for in experimental analyses. Conversely, failing to either correct for these cuts or provide a robust estimate of the associated uncertainties would limit the power of $\mathcal{A}_{FB}$ measurements for extracting non-perturbative parameters or testing lepton-flavor universality. Ideally, a future analysis will not employ transverse-momentum requirements but a cut on the lepton energy of 400 MeV in combination with a cut on $q^2$ of 3 GeV$^2$.

Simulations of $\mathcal{A}_{FB}$ with `Sherpa` including final-state radiation show that FSR effects are described well by using `PHOTOS` for soft photons and combining hard photons and electrons within small cones. For $q^2 > 3 \, \text{GeV}^2$ residual effects of hard radiation on $\mathcal{A}_{FB}$ are below 1%. Lowering the $q^2$ requirement further, however, introduces percent-level differences between electrons and muons.

With a measurement of $\mathcal{A}_{FB}$ in inclusive semileptonic $B$ meson decays within reach at Belle II, theoretical predictions must be improved commensurately to extract non-perturbative parameters in the HQE as precisely as possible. Corrections in the HQE of $\mathcal{O}\left(1/m_b^4\right)$ and $\mathcal{O}(\alpha_s)$ corrections must be calculated before a measurement of $\mathcal{A}_{FB}$ can be included in the global fit for $|V_{cb}|$. Both corrections can in principle be obtained along the same lines as higher-order corrections for the total rate or moments of distributions.

Further, dedicated studies of effective operators beyond the standard left-handed current are needed to assess the power of $\mathcal{A}_{FB}$ to constain physics beyond the Standard Model. Finally, it would be interesting to extend the analysis in this work in order to quantify the impact of the muon mass on the angular distribution, and to study the feasibility of measuring $\mathcal{A}_{FB}$ in $B \to X \tau \nu_\tau$ decays.

## Acknowledgments

FH is grateful to Raynette van Tonder for encouraging this work, numerous discussions on inclusive semileptonic *B* decays and the Belle II detector. FH thanks Manca Mrvar for pointing out Ref. [52] and communications regarding lepton identification requirements, as well as Stefan Hoeche, Hank Lamm and Ruth Van de Water for carefully reading and commenting on the manuscript.

FH acknowledges support by the Alexander von Humboldt foundation. This document was prepared using the resources of the Fermi National Accelerator Laboratory (Fermilab), a U.S. Department of Energy, Office of Science, HEP User Facility. Fermilab is managed by Fermi Research Alliance, LLC (FRA), acting under Contract No. DE-AC02-07CH11359.

## A    Analytic results for differences of partial moments

The differences of partial moments discussed in section 2.4 are given by:

$$
\begin{aligned}
\langle q^2 \rangle_- =& \frac{m_b^2}{10} \left( 1 - 27\rho + 160\rho^{3/2} - \rho^2(566 - 12\ln\rho) \right) \\
&+ \hat{\mu}_\pi^2 \left( -1 + 8\sqrt{\rho} - 36\rho + 120\rho^{3/2} - \rho^2(358 + 12\ln\rho) \right) \\
&+ \frac{\hat{\mu}_G^2}{5} \left( -8 + 40\sqrt{\rho} - 130\rho + 360\rho^{3/2} - \rho^2(1142 + 84\ln\rho) \right) \\
&+ \frac{\hat{\rho}_{LS}^3}{15 m_b} \left( -11 + 110\rho - 680\rho^{3/2} + \rho^2(2236 - 168\ln\rho) \right) \\
&+ \frac{\hat{\rho}_D^3}{15 m_b} \left( -\frac{91}{2} - 12\ln\rho + 160\sqrt{\rho} - 3\rho(72 - 76\ln\rho) - 15\rho^{3/2}(88 + 128\ln\rho) \right. \\
&\qquad\qquad \left. + \rho^2(8305 + 7830\ln\rho - 288\ln^2\rho) \right) + \mathcal{O}\left( 1/m_b^4, \rho^{5/2} \right),
\end{aligned}
\tag{A.1}
$$

$$
\begin{aligned}
\langle E_l \rangle_- =& \frac{m_b}{80} \left( 13 - 121\rho + 640\rho^{3/2} - \rho^2(1718 - 156\ln\rho) \right) \\
&+ \frac{\hat{\mu}_\pi^2}{480 m_b} \left( -181 + 1920\sqrt{\rho} - 8923\rho + 30080\rho^{3/2} - \rho^2(87434 + 2172\ln\rho) \right) \\
&+ \frac{\hat{\mu}_G^2}{480 m_b} \left( -523 + 1920\sqrt{\rho} - 5685\rho + 14720\rho^{3/2} - \rho^2(46302 + 4404\ln\rho) \right) \\
&+ \frac{\hat{\rho}_D^3}{480 m_b^2} \left( -2533 - 624\ln\rho + 5120\sqrt{\rho} - \rho(14603 - 816\ln\rho) \right. \\
&\qquad\qquad \left. - \rho^{3/2}(8320 + 30720\ln\rho) + \rho^2(4302 + 46116\ln\rho - 14976\ln\rho^2) \right) \\
&+ \mathcal{O}\left( 1/m_b^4, \rho^{5/2} \right)
\end{aligned}
\tag{A.2}
$$

and

$$
\begin{aligned}
\left\langle M_X^2 \right\rangle_- - M_B^2 \mathcal{A}_{FB} - \left\langle q^2 \right\rangle_- = M_b m_b \Bigg[ & \frac{1}{20} \left( -7 + 119\rho - 640\rho^{3/2} + \rho^2(2002 - 84\ln\rho) \right) \\
& + \frac{\hat{\mu}_\pi^2}{120 m_b^2} \left( 259 - 1920\sqrt{\rho} + 8197\rho - 26240\rho^{3/2} + \rho^2(77126 + 3108\ln\rho) \right) \\
& + \frac{\hat{\mu}_G^2}{120 m_b^2} \left( 497 - 1920\sqrt{\rho} + 5915\rho - 16000\rho^{3/2} + \rho^2(53578 + 4956\ln\rho) \right) \\
& + \frac{\hat{\rho}_{LS}^3}{120 m_b^3} \left( 231 - 791\rho + 7680\rho^{3/2} - \rho^2(20986 - 3780\ln\rho) \right) \\
& + \frac{\hat{\rho}_D^3}{120 m_b^3} \Big( (987 + 336\ln\rho - 1920\sqrt{\rho} + \rho(917 - 3024\ln\rho) \\
& + \rho^{3/2}(32640 + 30720\ln\rho) + \frac{\hat{\rho}_D^3}{120 m_b^3} \left( -\rho^2(136178 + 101724\ln\rho - 8064\ln^2\rho) \right) \Big) \\
& + \mathcal{O}\left( 1/m_b^4, \rho^{5/2} \right) \Bigg].
\end{aligned}
\tag{A.3}
$$

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
