# Peer review of "The forward-backward asymmetry and differences of partial moments in inclusive semileptonic B decays"

_SciPost Physics, doi:SciPost Phys. 14, 020 (2023)_

## Round 3 · Referee Report · Anonymous · 2022-9-9

Strengths
1-novel computation within the subject matter
2-timely work, given that the ongoing Belle II experiment will be able to measure the newly predicted observable
Weaknesses
none
Report
The author provides predictions for an observable and its kinematic moments in inclusive $B\to X\ell\bar\nu$ decays that had not previously been predicted, albeit discussed, in the literature. The observable is accessible at ongoing HEP experiments, to wit: the Belle II experiment. To me, this already qualifies the manuscript as eligible for publication.
The author discusses very methodically the prospects of an experimental determination of the proposed observable, taking care to investigate potential e.m. effects including final state radiation. He also discusses connections between his findings on the lepton identification for the inclusive decay and published Belle II results on the reconstruction efficiency in the exclusive decay $B\to D^*\ell\bar\nu$. The latter are currently under critical discussion due to a significant breaking $e/\mu$ universality and the manuscript might provide insight to understand this result, thereby increasing the relevance of this manuscript even further.
The manuscript is well written and mostly self contained. In fact, I would argue that it could have been reduced in length by a few pages; however, this does not enter into my recommendation.
Requested changes
none
Author: Florian Herren on 2022-11-08 [id 2998]
(in reply to Report 1 on 2022-09-09)I thank the referee for this report.
I have chosen not to reduce the length of the manuscript to keep it self-contained and to not sacrifice clarity.

---

## Editorial Decision

published